# A non-pyrophoric precursor for the low temperature deposition of metallic aluminium

Erica N. Faria, Samuel P. Douglas, Shreya Mrig, Leonardo Santoni ⓘ, Adam J. Clancy ⓘ, Daniel W. N. Wilson ⓘ & Caroline E. Knapp ⓘ ✉

The development of microelectronics prompts a search for precursors that can deposit conductive features. There is scarce research on Al as it is normally deposited using pyrophoric $AlH_3$ etherates/aminates. Ligands can impart increased stability while maintaining the ability to deposit target materials. Accordingly, we have engineered an aluminium complex that can undergo conversion to Al(0) at 100 °C. Our multi-step synthetic design features β-ketoiminate compounds, [Al(R-ketoiminate)$_2$Cl] (R = Me, Et, $^i$Pr, Ph and Mes, **1-5**) as starting materials to obtain aluminium hydride complexes: the polymeric amidoalane Li[AlH$_2$($^i$Pr-Hacnac)AlH$_3$]$_n$ (**6**) and the imidoalane cluster [AlH$_2$AlH$_2$(N-Mes)$_3$(AlH$_2$ · Li(Et$_2$O)$_2$)$_2$] (**8**). Decomposition of **8** into aluminium metal is observed when heated under vacuum at 100 °C and is confirmed by XRD, TEM, XPS. Deposition of a highly conductive film of Al is achieved from **8** after three weeks under nitrogen at room temperature. This represents a route to metallic aluminium involving non-pyrophoric precursors at low temperatures.

Modern electronics incorporating conductive metal tracks are integral to various aspects of modern life, encompassing applications such as light-emitting diodes[1,2], solar cells[3], biosensors[4,5], and wearable electronics[6–8]. Although technologies for large-scale flexible electronics manufacturing are already in place, such as inkjet printing, the metal inks currently employed present significant challenges. Specifically, the patterning of molten metals and sintering of printed metal (nano)particles are incompatible with cost-effective, flexible substrates such as plastics or paper[9,10]. This incompatibility arises partly due to the high melting points of metals, often exceeding 1000 °C, far surpassing the deformation/degradation temperatures of many plastic, paper, or fabric materials, typically in the range of 100–200 °C. A promising solution to these limitations lies in the development of designer inks containing small metal-containing molecules that can be thermally activated at sub-200 °C temperatures.

Aluminium is an earth-abundant, low-cost metal with conductivity comparable to silver and copper, making it an ideal metal for use in electronics[10]. However, its application is limited due to a lack of suitable precursors and their propensity to form nonconducting aluminium oxide[11,12]. Simple binary aluminium hydrides, known as alanes[13,14], are theoretically ideal precursors for the deposition of metallic aluminium features, as they contain highly reactive Al–H bonds and are, therefore prone to decomposition at lower temperatures[11,15]. It has been shown that alanes are able to deposit aluminium however, such precursor compounds are remarkably oxygen-sensitive, pyrophoric reagents that require handling strictly under dry and inert atmosphere. These characteristics render decomposition temperatures below 100 °C but make their use as precursors in real-world applications challenging and dangerous[16].

Compounds containing Al–N bonds have been the subject of attention owing to the intriguing structures they can adopt[17] as well as their potential as precursors for binary materials[18]. The Al–N bond is highly polarised, with Pauling electronegativity of 1.61 for Al and 3.04 for N[19] and tend to form cyclic structures with the degree of oligomerization dependant on the steric bulk of the organic substituents both at the nitrogen and aluminium atoms[20,21]. Few examples of

Department of Chemistry, University College London, London, UK. ✉e-mail: caroline.knapp@ucl.ac.uk

aluminium complexes containing both conjugated five-membered ring chelates and halide ligands have been reported[22–24]. In this class, aluminium complexes featuring two β-ketoiminate ligands and one chloride ligand of the type [AlL₂Cl] have been prepared (Fig. 1)[24–26]. These Al–N and Al–O containing β-ketoiminates have been used for CVD deposition previously[27,28], and other Metal–N complexes are known precursors for low temperature metal printing of other highly conductive metals such as silver[10].

Herein, we report a non-pyrophoric precursor to aluminium metal and explain the conceptual design through isolation of complexes of the type [Al(R-acnac)₂Cl]: careful ligand design allowed molecules to exhibit increased stability towards moisture and air. Thermal treatment of the leading complex, an amine-supported tetra-aluminium hydride (**8**), at 100 °C overnight results in the formation of aluminium metal. Conversion also occurs when heating to 150 °C for 1 h, alternatively a highly conductive aluminium film on glass is produced when stored under nitrogen at room temperature for 3 weeks.

## Results and discussion

The ideal Al precursor is one which degrades to give a large fraction of Al within a reasonable temperature window (80–200 °C), while being safe to handle. To this end, the species should have a high Al atomic weight loading and degradation-inducing weak bonds (e.g., Al–H) balanced by stability-inducing stronger-bonding ligands and bulky sterics, the latter of which often conflicts with the Al fraction. Oxygen content should also be minimised to discourage the parasitic formation of Al₂O₃. An aluminium hydride stabilised by a single β-ketoiminate provides a promising conceptual starting point, as these ligands are known to stabilise Al, contain an Al–N bonds to reduce oxygen content (under the 1.5 alumina O:Al ratio for 1 ligand), have imine R-groups to systematically control ligand steric bulk to dictate thermal stability, and have been reported previously as ligands to useful precursors[27,28].

Therefore, as a starting point, the complexes used in this study were prepared from a range of bis(β-ketoiminate) aluminium chloride compounds (Fig. 1; **1-5**). In a given reaction, two equivalents of deprotonated β-ketoiminate were reacted with AlCl₃ in toluene at

−78 °C, resulting in the desired bis(β-ketoiminate) aluminium chloride. The resulting compounds were fully characterised using standard techniques (¹H and ¹³C{¹H} NMR spectroscopy, Single Crystal X-Ray Diffraction, Mass Spectrometry, and Elemental Analysis). The ¹H NMR spectra of **1-5** display loss of the NH proton resonance of the ligand and all peaks are shifted with respect to the associated free ligand.

The solid-state structures of **1-5** confirm their identity as [AlL₂Cl] complexes. In all cases, the complexes are monomers containing two chelating bidentate β-ketoiminate ligands and one chloride bound to the aluminium atom, in a mildly distorted trigonal bipyramidal arrangement around the metal (τ – structural parameter – values of **1**: 0.87; **2**: 0.95; **3**: 0.95; **4**: 0.86; **5**: 1). For **1-4**, the Cl and O atoms occupy the equatorial positions while the N atoms occupy the axial positions. This motif is preferred for ligands with less bulky R groups, while the steric bulk of the mesityl group in **5** results in reversal of the axial and equatorial arrangement (See SI sections 1.1–1.5). The two AlOC₃N six-membered rings are angling away from the large Cl atom, with the N(1)–Al(1)–Cl(1) and N(2)–Al(1)–Cl(1) angles of 90.73(4)-120.09(4)°and 90.12(4)-120.09(3)°, respectively. There is also distortion between the axial and equatorial positions caused by steric constraints of the ligand, displayed by the pincer angles O(1)–Al(1)–N(1) and O(2)–Al(1)–N(2) of 90.16(4)-92.03(5)° for both. There are distortions between equatorial atoms, with O(1)–Al(1)–Cl(1) and O(2)–Al(1)–Cl(1) angles of 89.88(4)-120.61(4)° and 89.88(4)-119.37(4)°, respectively, which deviates from the expected 120° angle, most likely caused by the presence of the large electronegative Cl atom. As expected, the Al–O bond lengths of 1.7718(11)-1.8286(12) Å are shorter compared to the Al–N bond lengths of 1.9718(13)-2.0377(11) Å. The Al–Cl bonds of 2.1896(7)-2.2197(5) Å are comparable with the examples of [AlL₂Cl] in the literature[24–26].

The bulkiness of the organic ligands has a pronounced impact on the thermal stability of the compounds as measured by thermogravimetric analysis (TGA), with mass loss initiating below 200 °C for **1-4** and at 278 °C for the bulk-mesityl containing **5** (Section 3 of SI). The formation of the hydrides through substitution of the chlorine to introduce more reactive Al–H bonds was expected to lower the degradation temperature into the desired window (<200 °C).

**Fig. 1 | Synthesis of complexes 1-8:** i) Reaction of [Al(ⁱPr-acnac)₂Cl] with 3 equivalents of LiAlH₄ in diethyl ether at −78 °C to room temperature to form Li[AlH₂(ⁱPr-Hacnac) AlH₃]ₙ (**6**). Crystalline yield: 25 mg (27%); ii) NMR scale reaction of **6** with 1.04 equivalents of 12-crown-4 to obtain the monomeric [Li(12-crown-4)][AlH₂(ⁱPr-Hacnac) AlH₃] (**7**); iii) Reaction of [Al(Mes-acnac)₂Cl] with 3 equivalents of LiAlH₄ in diethyl ether at −78 °C to room temperature to obtain **8**. Crystalline yield: 72.4 mg (6%); iv) Reaction of 4 equivalents of LiAlH₄ with 3 equivalents of NH₂–Mes in diethyl ether at −78 °C to room temperature to alternatively obtain **8**. Yield: 0.352 g (34%).

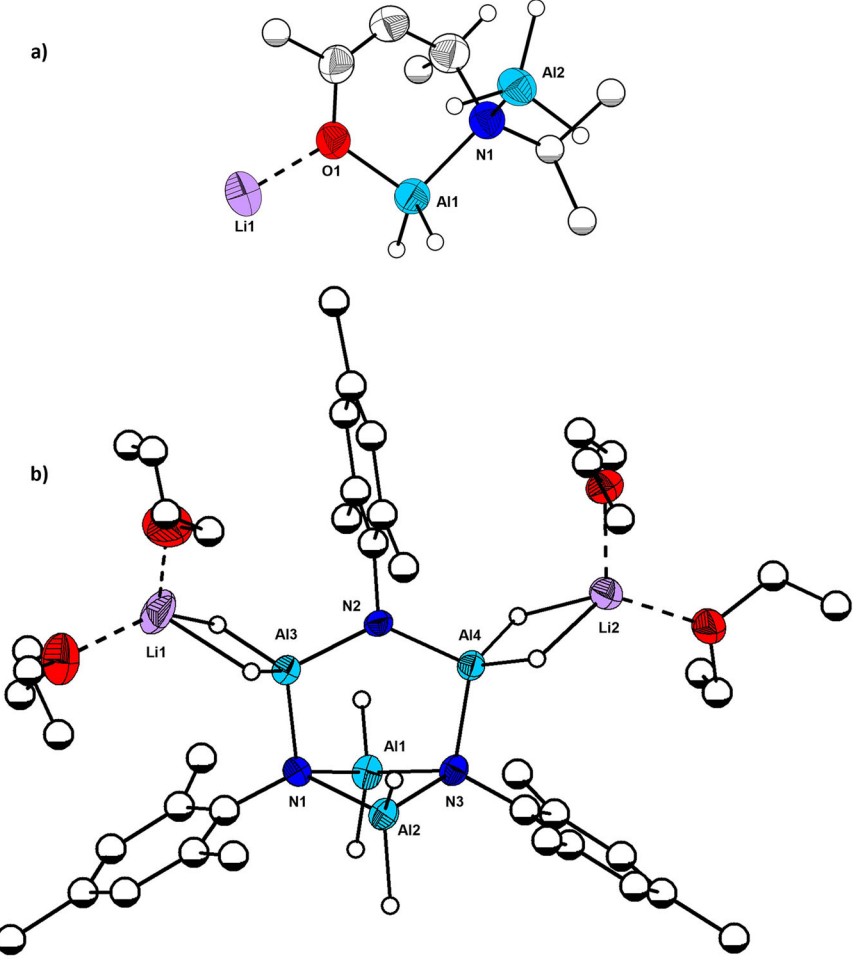

**Fig. 2 | Molecular structures of 6 and 8. a** Repeating unit of Li[AlH$_2$($^i$Pr-Hacnac)AlH$_3$]$_n$ (**6**); **b** Molecular structure of the cluster [AlH$_2$AlH$_2$(N−Mes)$_3$(AlH$_2$ · Li(Et$_2$O)$_2$)$_2$] (**8**). *The ellipsoids are shown as 50% probability. Most hydrogen atoms are omitted, and some carbon atoms are shown as spheres of arbitrary radius.* Selected bond lengths (Å) of **6**: O1−Li1: 1.914(9); O1−Al1: 1.786(4); O1−C1: 1.376(6);

C1−C2: 1.327(7); C2−C3: 1.502(7); C3−N1: 1.521(7); N1−Al1: 1.937(4); N1−Al2: 1.969(4). Selected bond lengths (Å) and angles (deg) of **8**: Al1−N1: 1.963(4); Al1−N3: 1.963(4); Al2−N1: 1.977(4); Al2−N3: 1.956(4); N1−Al3: 1.902(4); Al3−N2: 1.815(4); N2−Al4: 1.816(4); Al4−N3: 1.903(4); Al3−N2−Al4: 128.7(2); Al3−N2−C10: 111.4(3); Al4−N2−C10 119.9(3).

Conventional hydride sources such as LiBHEt$_3$, CaH$_2$, ($^i$Bu$_2$AlH)$_2$ and LiAlH$_4$ were used to attempt the conversion of the chloride into a hydride and form [Al(R-acnac)$_2$H]. Of these hydride sources, only reactions with three equivalents of LiAlH$_4$ yielded promising results. From compounds **1-5**, single crystals of **6** and **8** [from Al(R-acnac)$_2$Cl] (R = $^i$Pr (**3**) and Mes (**5**)]) were obtained, most likely owing to their higher steric bulk that would be able to better stabilise the reactive Al−H bond.

The reaction of **3** with 3 equivalents of LiAlH$_4$ in diethyl ether gave rise to the polymeric Li[AlH$_2$($^i$Pr-Hacnac)AlH$_3$]$_n$ structure (**6**) (Fig. 2 and Fig. S2a). The solid-state structure has a repeating unit featuring one chelating bidentate β-ketoiminate ligand bonding to one aluminium atom via both oxygen and nitrogen atoms, with the Al1−O1 bond (1.786(4) Å) shorter than the Al1−N1 (1.937(4) Å) bond. The backbone of the ligand has been protonated, resulting in loss of delocalisation of its π-electronic system due to the presence of the newly formed sp$^3$ hybridised C3 (Fig. 1), that is now chiral, as evidenced both by the deviation of planarity of the ligand as well as by the pattern of bond lengths in the ligand backbone that now indicate a localised C1−C2 double bond (O1−C1: 1.376(6) Å, C1−C2: 1.327(7) Å, C2−C3: 1.502(7) Å, C3−N1: 1.521(7) Å) when compared to the β-ketoiminate pro-ligand (O1−C1: 1.247(4) Å, C1−C2: 1.396(4) Å, C2−C3: 1.396(4) Å, C3−N1: 1.337(3) Å)[29]. The nitrogen atom is coordinated to a second aluminium atom (Al2−N1: 1.969(4) Å) and the oxygen is also coordinated to a

lithium atom (Li1−O1: 1.914(9) Å). The extended structure is formed through bridging hydride ligands between the electrophilic Al and Li sites, resulting in a zig-zag coordination polymer (Fig. S2b).

As a result of the backbone protonation, the ligand exhibits a −2 charge. While, to our knowledge, such a ligand reduction has not been reported for β-ketoiminate complexes, Uhl and co-workers have reported a similar transformation in a related bis(β-diketiminate) aluminium complex[30]. The exact formation mechanism of **6** remains unclear, although the product suggests either a step involving hydro-alumination across the C−N double bond of the ligand[31,32], or hydride transfer to the electrophilic carbon C3 and, in turn, forming the unusual dianionic amido-enolate ligand. The monomeric form of **6** can be obtained by addition of a slight excess of the sequestering agent 12-crown-4 to a suspension of **6**, yielding **7** (Fig. 1 and Fig S3.). A more detailed discussion can be found in the supporting information (Section 1.7).

In contrast to the isopropyl-based **3**, addition of 3 equivalents of LiAlH$_4$ to the mesityl complex **5** results in destruction of the ligand to form the tetra-aluminium cluster [AlH$_2$AlH$_2$(N−Mes)$_3$(AlH$_2$ · Li(Et$_2$O)$_2$)$_2$] (**8**; Fig. 2). The solid-state structure reveals that only the N−Mes moiety of the β-ketoiminate ligand is present after complexation, with the nitrogen atoms forming part of the cyclic core that encompasses three nitrogen atoms alternating with four aluminium atoms. There are a few examples of related β-diketiminate ligands

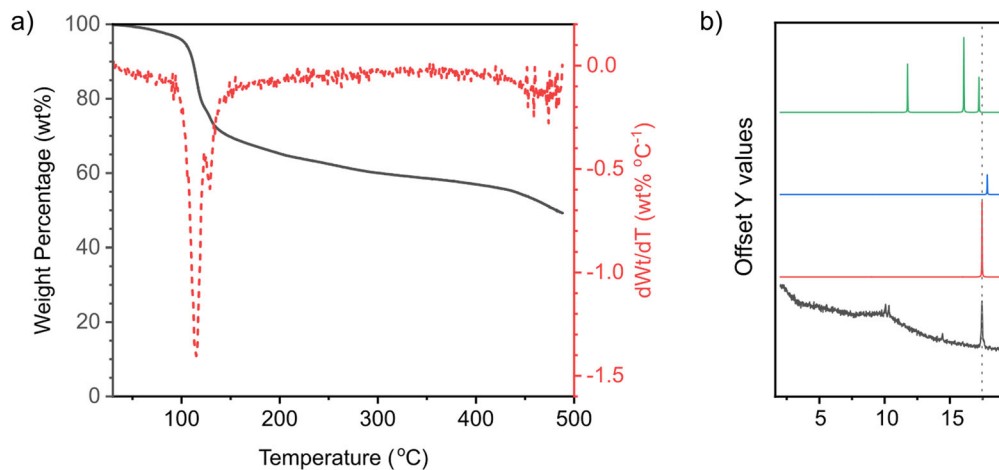

**Fig. 3 | Thermogravimetric and powder diffraction data of 8. a** Overlapping thermogram (full grey line) and derivative (dashed red line) of compound **8**; **b** Overlapping PXRD (measured with Mo kα) of aluminium deposited from

**8** (grey), reference of Al (red), reference of AlN (blue) and reference of Al₂O₃ (green). Reference patterns obtained from the Crystallography Open Database (COD).

undergoing degradation by hydride sources described in the literature[33–36]. Such degradation involves hydride migration onto the β-carbon followed by C–N bond cleavage, therefore, we envision a similar mechanism is involved in the formation of **8**. Additional investigations into the reaction of LiAlH₄ with aniline and various substituted β-ketoiminates resulted in intractable mixtures (SI section 2.3).

In the structure of **8** (Fig. 2), N2 resides in a distorted trigonal planar geometry environment, with bond angles at Al3–N2–Al4, Al3–N2–C10 and Al4–N2–C10 of 128.7(2)°, 111.4(3)° and 119.9(3)° respectively (sum of the bond angles around N2: 360°). Additionally, the Al3–N2 and N2–Al4 bond lengths of 1.815(4) and 1.816(4) are shorter than the other Al–N bonds present in the molecule and shorter than the sum of the covalent radii of the individual atoms ($r_N$ = 0.71 Å, $r_{Al}$ = 1.21 Å; $r_{sum}$ = 1.91 Å)[37], indicating that either delocalisation of the nitrogen lone pair into vacant orbitals on the aluminium, presumably $\sigma^*$, or increased localisation of charge at the three-coordinate nitrogen results in a stronger ionic N–Al interaction in comparison to the other two four coordinate N1 and N3. These Al–N bond lengths are consistent with other cyclic Al–N bonds present in the literature[20], N1 and N3 adopt a tetrahedral geometry as they are each bound to three aluminium atoms instead of two, creating a butterfly-like fragment consisting of N1Al1N3Al2, with the distance between the Al1 and Al2 atoms of 2.684 Å. The Al1–N1 (1.963(4) Å), Al1–N3 (1.963(4) Å), Al2–N1 (1.977(4) Å) and Al2–N3 (1.956(4) Å) bond lengths are longer than N1–Al3 (1.902(4) Å) and Al4–N3 (1.903(4) Å), most likely due to the geometry constraint imposed by the presence of a fourth Al atom in the N₃Al₄ core[38]. These values are in agreement with other cyclic Al–N bonds exhibiting both N and Al atoms in tetrahedral geometries present in the literature[39–42]. Each molecule of **8** also contains two hydride bridged Al–Li bonds with each four-coordinated lithium further stabilised by two ether molecules.

Density functional theory calculations were performed to better understand the bonding in **8**. Gas phase geometry optimisations were performed using the PBE0 functional and the def2-TZVP basis set with Grimme's D3 dispersion correction used on all atoms. The gas phase optimised structure reveals an Al1–Al2 distance of 2.730 Å, and slightly inequivalent Li–Al3/4 distances of 2.603/2.613 Å, in good agreement with the solid-state structure. Natural bond order analysis found no bonding interaction between Al1 and Al2, implying that the unusually short Al–Al distance is due to the constraint imposed by the heterocyclic ring. Analysis of the Mulliken charges indicate that the $\mu^2$-H are

more hydridic than the terminal Al–H (average partial charge of –0.28 for $\mu_2$-H vs –0.13 for Al–H), likely due to their proximity to the hard, cationic Li atoms, which are more electropositive in comparison to Al (average partial charge of +0.44 for Li, +0.06 for Al3/4).

Given that the β-ketoiminate ligand that was present in the starting material [Al(Mes-acnac)₂Cl] is not present in the final structure of **8**, the rational synthesis of the aluminium cluster was attempted by reacting a 3:4 ratio of 2,4,6-trimethylaniline and LiAlH₄ in diethyl ether (Scheme S2). The reaction was confirmed by ¹H NMR spectroscopy and allowed for isolation **8** in 34% yield, allowing for a more atom and time efficient path to obtaining the aluminium cluster (**8**). Notably, stirring of **8** in toluene at 1.33 mg/mL leads to formation of a suspension which precipitates slowly over several days, highlighting its potential future use in printing applications.

Excitingly, TGA performed on the polymeric **6** and monomeric **8** displayed lower onset temperatures for mass loss in comparison to **1-5**, with **8** initiating decomposition already at 50 °C, which is a lower temperature than previously reported for pyrophoric alanes, or indeed any aluminium precursor[10,43]. The first derivative curve for **6** indicate two main mass-loss events, associated with ~62% of the total mass (Fig. S37). The first derivative curve for **8** indicates two mass-loss events, associated with ~49% of the total mass (Fig. 3a). Variable temperature powder X-ray diffraction (PXRD) indicates that the material undergoes gradual phase change at lower temperatures, before transitioning through multiple distinct (currently unidentified) crystalline phases between 80 and 100 °C, forming a material stable by 110 °C which remains stable up to 200 °C. While (partially) crystalline, the material was not metallic aluminium, attributed to retention of ligand atoms (Fig. S42).

To circumnavigate this theorised issue, **8** was heated at 150 °C under dynamic vacuum for 1 h, to remove degrading volatile species before they could interact with the reactive Al-containing intermediate. Preliminary optimisation experiments also show conversion to aluminium at 100 °C overnight. In both cases, the mixture formed a grey solid which was confirmed to contain metallic aluminium by PXRD (Fig. 3b). Transmission electron microscopy (TEM, Fig. 4a-c) of the material indicates that the powder consists of ~1–10 μm species each consisting of smaller sintered regions, with selected area diffraction confirming that the material contained polycrystalline aluminium (Fig. 4d). Additionally, when left for 3 weeks at room temperature a highly conductive Al metallic film could be obtained. The thin film on glass thickness was measured using cross-sectional SEM to measure ~1 μm (Fig. 4d), the film exhibited a measured sheet resistivity of

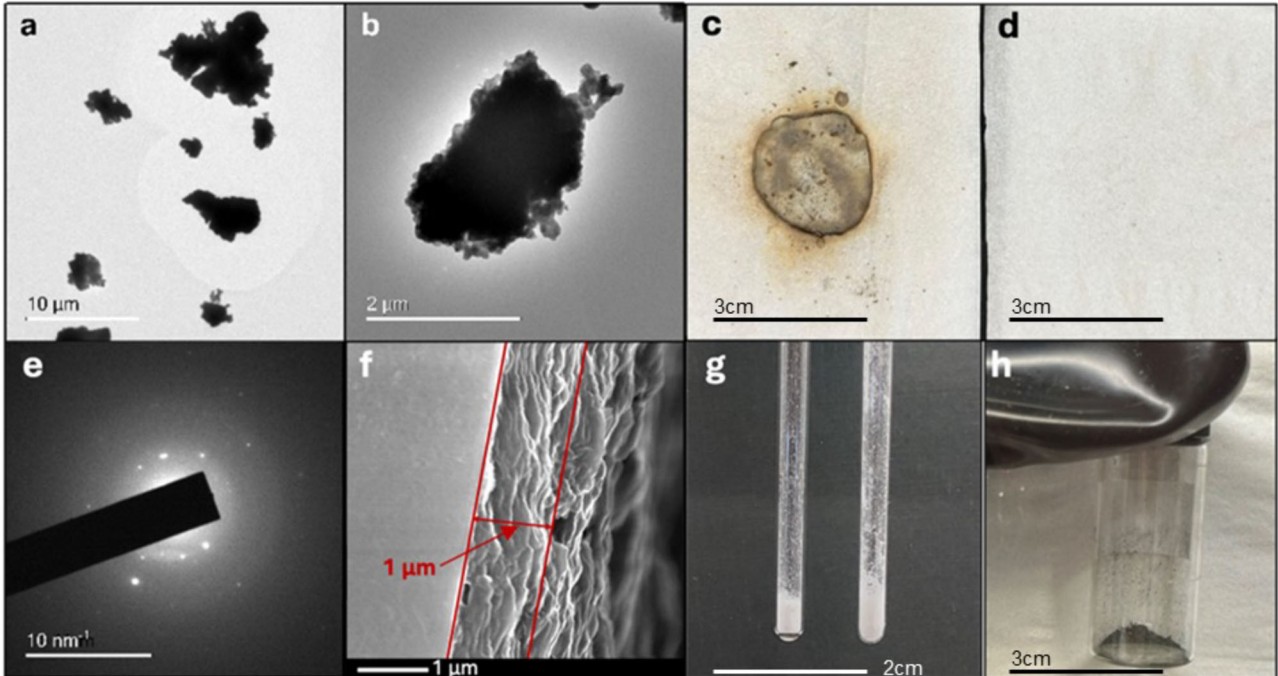

**Fig. 4 | Materials characterisation and images of 8. a, b** TEM micrographs of dispersed Al powder from **8** heated under vacuum; **c** tissue after addition of hexane solution of DMEAA; **d** and tissue after addition of a toluene solution of cluster **8**; **e** selected area electron diffraction (SAED) of dispersed Al powder from **8** heated under vacuum; **f** Side-on SEM of aluminium film on glass, highlighting thickness of film; **g** precursor **8** before heating and **h** grey powder containing metallic aluminium.

$4.34 \pm 0.71\,\text{k}\Omega/\text{sq}$ which is comparable with Al films obtained using other precursors and deposition techniques[43], grazing incidence (GI) XRD pattern confirmed the presence of aluminium (Fig. S43, SI section 4.3). Notably and importantly, in contrast to other aluminium metal precursors such as alane, in this work, the precursor is not pyrophoric. While exposure to air decomposes **8** (Section 6.2 in SI) it does not ignite and leads to slow change of colour alone. Additionally, this route to Al(0) is free of exogenous reductant, instead the ligand is oxidised to aid formation of metallic aluminium. Similarly, while syringing a hexane solution of dimethylamine alane (DMEAA) onto a tissue paper in air leads to ignition of the paper, a comparable test with a toluene suspension of **8** leads to no visible effects (Fig. 4). Videos of this test for each compound can be found in SI and highlight the superior user-friendly character of **8** (Section 6 in SI, Supplementary Movies 1 and 2).

This work presents a multi-step design route toward a bespoke molecule resulting in a nonpyrophoric aluminium compound which can deliver the metal at 150 °C in 1 h: cluster **8** holds ultimate potential to be used as a precursor for the deposition of metallic aluminium. Whilst this is still undergoing optimisation, preliminary experiments reveal conversion to also be accessible at 100 °C overnight giving rise to Al, as confirmed by PXRD diffraction experiments. A longer time-frame of 3 weeks at room temperature formed a highly conductive Al film on glass with a thickness of 1 μm, with a measured sheet resistivity of $4.34 \pm 0.71\,\text{k}\Omega/\text{sq}$ comparable to other films in the literature[43].

A family of compounds containing two β-ketoiminate ligands and one chlorine co-ligand bound to an Al atom have been synthesised via the reaction between one equivalent of $AlCl_3$ and two equivalents of [Li(R-acnac)], to give five [Al(R-acnac)$_2$Cl] products, **1-5**, in good yield. The products include the first example where the *N*-substituent is an alkyl group. Each complex adopts a distorted trigonal bipyramidal geometry around the Al metal, with the Cl atom in the equatorial position and the two β-ketoimine ligands binding in a bidentate mode where the chelating rings are delocalised. When less bulky *N*-substituents are used, the oxygen atoms occupy the equatorial positions and, when bulkier R groups are used (R = Mes), such as in complex **5**, the nitrogen atoms of the ligand occupy the equatorial positions instead.

In order to probe this set of complexes suitability as precursors, each compound was thermally analysed via TGA. Compounds generally showed reasonable decomposition properties, with the best potential precursor being [Al(Et-acnac)$_2$Cl] (**2**), displaying good mass loss properties and a low onset decomposition temperature (<200 °C).

The reaction of compounds **3** (R = $^i$Pr) and **5** (R = Mes), with three equivalents of LiAlH$_4$ resulted in the formation of two completely different and novel aluminium hydride structures, namely the polymer **6** and the cluster **8**. In **6**, the β-ketoiminate ligand has had a hydride added, therefore losing delocalisation of its π-electronic system and exhibiting a localised double bond instead, causing the ligand to adopt the unusual −2 charge, only previously described in one case with the closely related β-diketiminate ligand[30]. In cluster **8**, the ligand also undergoes an unusual decomposition leading to a product containing only the amine moiety from the initial ligand, a phenomenon that has only been seen in rare occasions also with β-diketiminate ligands[33–36]. Compound **8** exhibits both imido- and an amino-alane characteristics due to the presence of both trigonal and tetrahedral nitrogen atoms bound to aluminium atoms.

## Methods
### General experimental details
All reactions and product manipulations were performed using standard Schlenk-line and glovebox techniques, both using N$_2$ as the inert gas. All glassware used in air-sensitive reactions were flamed dried prior to use. All reagents were procured from Sigma Aldrich/Merck and used as received unless otherwise stated. All non-deuterated solvents were stored under N$_2$ in gas-tight ampoules over activated molecular sieves (3 or 4 Å)[5]. Deuterated solvents were obtained from Cambridge Isotope Laboratories or Sigma Aldrich/Merck and were degassed before use and stored over 3 Å molecular sieves. Amines were used as received for ligand synthesis and the β-ketoiminate ligands were synthesised according to previously reported literature from the group[6]. LiAlH$_4$ powder, reagent grade 95%, was procured from Sigma Aldrich and purified by dissolution and recrystallisation from diethyl ether using a literature procedure[7]. Further details on NMR, Elemental

Analysis, Mass Spectrometry, TGA, IR, single crystal X-ray diffraction (SCXRD) and powder XRD, SEM and conductivity measurements are included in SI. Details of all synthetic procedures, including routes to compounds **1-5** complete with all crystallography and spectra are included in the SI, along with thermal decomposition profiles, XRD, VT-XRD, XPS and TEM.

### Synthesis of Li[AlH₂(ⁱPr·acnacH)AlH₃]ₙ (6)

Diethyl ether (5 mL) was added of [Al(ⁱPr-acnac)₂Cl] (0.153 mg, 0.447 mmol). In another Schlenk flask, diethyl ether (10 mL) was added to LiAlH₄ (50.9 mg, 1.34 mmol) to form a cloudy colourless solution. The LiAlH₄ was dropwise added to the [Al(ⁱPr-acnac)₂Cl] mixture at −78 °C. The mixture was stirred at low temperature for 1 h, then the cold bath was removed and the mixture stirred for further 1 h. The colourless solution was filtered and the volatiles removed under reduced pressure to obtain a white solid. The solid was redissolved in ether and layered with hexane to obtain colourless crystals of the product suitable for SCXRD. Crystalline yield: 25 mg (27%). **IR (cm⁻¹):** 2969, 1834, 1738 [$v$(Al−H)], 1658, 1373, 1314, 957, 887, 697, 613. **CHN:** Found (Calcd.) for $C_8H_{18}Al_2LiNO$: 44.91(46.84), 9.25(8.84), 6.34(6.83).

### Synthesis of [AlH₂AlH₂(N·Mes)₃(AlH₂ · Li(Et₂O)₂] (8)

Diethyl ether (40 mL) was added to [Al(Mes·acnac)₂Cl] (0.725 g, 1.46 mmol) to form a cloudy solution. In another Schlenk flask, diethyl ether (50 mL) was added to LiAlH₄ (166.7 mg, 4.39 mmol). The cloudy solution of LiAlH₄ was dropwise added to [Al(Mes·acnac)₂Cl] at −78 °C. The mixture was stirred at low temperature for 1 h, then the cold bath was removed and the mixture stirred for further 1.5 h. The colourless solution was filtered and the volatiles removed under reduced pressure to form a foamy sticky solid that can be broken into a white powder when dipped into liquid nitrogen. The solid was redissolved in ether to obtain colourless crystals of the product at room temperature suitable for SCXRD. Crystalline yield: 72.4 mg (6 %). **¹H NMR** ($C_6D_6$, 500 MHz) δ 6.78 (1H, d, Ar−H), 6.74 (1H, d, Ar−H), 4.47 (s, Al$H_2$ · Et₂O), 4.22 (br s, Al$H_2$−NMes), 3.18 (q, O(C$H_2$CH₃)₂), 2.81 (3H, s, *o*-C$H_3$), 2.77 (3H, s, *o*-C$H_3$), 2.10 (3H, s, *p*-C$H_3$), 1.04 (t, O(CH₂C$H_3$)₂). **¹³C{¹H} NMR** ($C_6D_6$, 126 MHz) δ 144.9 (C$_{Ar}$−N), 132.5, 132.07, 131.37, 131.27 (*o*-C$_{Ar}$ and *m*-C$_{Ar}$), 130.89 (*p*-C$_{Ar}$), 66.38 (O(CH₂CH₃)₂), 25.79 (*o*-C$H_3$), 22.44 (*o*-C$H_3$), 20.64 (*p*-C$H_3$), 15.63 (O(CH₂C$H_3$)₂). **⁷Li NMR** ($C_6D_6$, 194 MHz) δ 0.44 (bs) **²⁷Al NMR** ($C_6D_6$, 130 MHz) δ 70.92 (bs) **IR (cm⁻¹):** 2975, 2906, 1783 [$v$(Al−H)], 1654, 1662, 1472, 1198, 1136, 823, 738, 673. **CHN:** Found (Calcd.) for $C_{43}H_{81}Al_4Li_2N_3O_4$: 59.92(62.53), 8.78(9.89), 5.24(5.09).

### Alternative route to obtain [AlH₂AlH₂(N·Mes)₃(AlH₂ · Li(Et₂O)₂] (8)

2,4,6-trimethylaniline (NH₂-Mes) (0.5 mL, 3.70 mmol) was degassed prior use by the freeze, pump, thaw method. This was dissolved in diethyl ether (10 mL) in one Schlenk flask and added to another Schlenk flask containing LiAlH₄ (187 mg, 4.93 mmol) and diethyl ether (40 mL) at −78 °C. The mixture was stirred in the cold bath for 15 min, then the cold bath was removed and the mixture allowed to stir for further 40 min. The resulting colourless solution was filtered the solvent removed to obtain a white foam that was turned into a solid by dipping the flask under vacuum in liquid nitrogen. Colourless crystals were obtained from diethyl ether at room temperature. Yield: 0.352 g (34%).

## Data availability

The authors declare that the data supporting the findings of this study are available within the article and its Supplementary Information Files. All data are available from the corresponding author upon request. The X-ray crystallographic coordinates for structures reported in this study have been deposited at the Cambridge Crystallographic Data Centre (CCDC), under deposition numbers 2405550-2405557. This data can be obtained free of charge from The Cambridge Crystallographic Data Centre via www.ccdc.cam.ac.uk/data_request/cif.

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

## Acknowledgements

We thank Dr Jamie Gould for help with PXRD experiments and Dr Andrew Stuart for TEM. The EPSRC are thanked for funding this work (EP/V027611/1, EP/Y001877/1). A.J.C. would like to acknowledge the Ramsay Memorial Trust and The Royal Society for funding through the University Research Fellowship scheme (URF\R1\221476, RF\ERE\221017). D.W.N.W. thanks the Royal Commission for the Exhibition of 1851 for funding (RF2023DW).

## Author contributions

C.E.K. and E.N.F. conceptualised the project. E.N.F. and S.P.D. conducted the synthesis and characterisation of novel complexes. S.M., L.S., A.J.C. and D.W.N.W. aided with characterisations of complexes. DWNW performed the density functional theory calculations. C.E.K., E.N.F., D.W.N.W., and A.J.C. prepared the manuscript. All authors provided feedback on the manuscript.

## Competing interests

The authors declare no competing interests.
