## [Transparent Peer Review file · Nature Communications]

A Non-Pyrophoric Precursor for the Low Temperature Deposition of Metallic Aluminium

Corresponding Author: Dr Caroline Knapp

Version 0:

Reviewer comments:

Reviewer #1

(Remarks to the Author)

Comment on manuscript entitled "The First Non-Pyrophoric Precursor for the Low Temperature Deposition of Metallic Aluminium"

This manuscript has two parts.

The first part of this manuscript describes the synthesis and characterization of a series of penta-coordinated Al(III) complexes coordinated to β -ketiminato (acnac) ligands with the general formula $\text{Al}(\text{R-acnac})_2\text{Cl}$ (1-5). All of these complexes are satisfactorily characterized. The molecular structures of these complexes are reported as expected.

In the second part, the authors reacted all these aluminium precursors with various equivalents of lithium aluminium hydride to obtain suitable aluminium hydride precursors for elemental aluminum deposition. In line with the relatively poor thermal stability of the aluminium hydride complexes, aluminium hydride complexes synthesized with relatively bulky -R [ipr and Mes] groups are isolated and characterized (6, 7, and 8). The molecular structures of complexes 6, 7, and 8 are determined using single-crystal X-ray diffraction studies. Interestingly, complex 8 is formed via the complete release of the attached β -ketiminato groups. Further, looking at the β -ketiminato ligand-free structure of complex 8, the authors developed an easier and more convenient route by a simple reaction between mesityl-aniline and lithium aluminium hydride. Finally, the authors established, or attempted to establish, that complex 8 is air-stable and can decompose below 100°C to deposit elemental aluminium.

The manuscript is well written, and most of the claims are supported by experimental data.

My comments:

(1) The reason for the preferential use of β -ketiminato ligands is not described or justified anywhere in the manuscript. I would argue for the use of the well-known hydrazine and hydroxyl complexes of aluminium hydride for the above-stated purpose.

(2) Compound 8 is the most important molecule in this manuscript, as described by the authors, and it can be synthesized following a route (preferable) without the use of β -ketiminato ligands. Thus, the synthesis, characterization, and chemistry discussed in the major part of this manuscript have no relation to the actual goal of this work. It seems that the important molecule 8 can also be synthesized following a much longer and more tedious β -ketiminato route.

(3) I would suggest (with full respect) reporting the β -ketiminato chemistry in a different manuscript. Additionally, the authors should develop a new route for the synthesis of diverse precursors following the reaction between lithium aluminium hydride and diverse aniline/amine derivatives. Only then will the true potential of such aluminium complexes be understood in their ability to deposit metallic aluminium below 100°C. The underlying chemistry would be very important for the development of aluminium complexes for metallic aluminium deposition.

(4) The authors have carefully used the word "non-pyrophoric" in the title of the manuscript. However, in my opinion, molecule 8 may not catch fire upon contact with air but could decompose in solution if exposed to moist air for a relatively long time. The lithium ion coordination to the hydrogen atom associated with aluminium likely reduces the reactivity of 8 towards oxygen (I guess this may be stable under dry oxygen).

(5) All the characterization data support the formation of aluminium metal via the decomposition of 8, but this is a one-off example. It would only be conclusive through device design and measurement of conductivity potential.

Reviewer #2

(Remarks to the Author)

Overall, this is a succinct but detailed report on the discovery and synthesis of a novel aluminium metal precursor. It represents an excellent piece of well thought out and systematic inorganic synthesis, showcasing how an appreciation of the fundamentals of ligand design, with a careful and systematic synthesis strategy, can lead to discovery of a complex with ideal properties for a desired application. As such, I would like to strongly recommend the publication of this work in Nature Communications, (after some minor corrections).

In the introduction the authors make a clear case for the value of an air stable aluminium precursor that can be incorporated into an ink formulation for low temperature reduction to aluminium metal, and the challenge that this presents in terms of avoiding pyrophoric materials, or precursors that decompose to the more stable alumina. Although the ink formulation is the key focus for the authors, their precursors would also see potential usage in the key industrial technique of CVD.

In the results, the authors then discuss the design strategy, justifying why the system of a hydride stabilised by beta-ketoiminates is a likely candidate for the desired properties. They then report their systematic trial of different beta-ketoiminates (L) ligands, to form AlL_2Cl complexes, and then the results of the attempts to replace the chloride with a hydride by reaction with $LiAlH_4$. Comprehensive characterisation of all products by SXRD, TGA, NMR, MS and elemental analysis is provided, with a detailed discussion of the SXRD structures in the main paper, with comparison to suitable, related compounds in the literature, highlighting some unusual features of interest to main group chemists.

This careful approach to the analysis reveals the most significant finding is that 'compound 8' will decompose under vacuum to aluminium metal at temperatures from 100 to 150 C, and that this precursor is non-pyrophoric. This is a significant innovation, and the videos provided in the SI are a nice touch to highlight this difference compared to dimethyl alane.

The supplementary information, provides all necessary information for the experiments to be replicated, and further and more specific analysis of each sample, including figures of the NMR IR spectra, and crystal structures confirming the details provided in the main paper.

As stated above, my recommendation is to publish, but I would like the following points and corrections to be addressed:

- 1) In the abstract the authors state that the aluminium complex undergoes 'reductant free' conversion. I understand that they mean there is no specific reducing agent used, but as the result is metallic aluminium, there must be a reductant to meet the rules of redox chemistry and electron bookkeeping, even if it is just the one of the ligands.
- 2) There is a spelling mistake on 'ellipsoids' in figure caption 2
- 3) On p7 in "most likely due to the geometry constrain imposed by the presence of a fourth Al atom in the N_3Al_4 core" the word constrain should probably read constraint
- 4) In the conclusions should the word be losing instead of loosing?
- 5) In the supplementary information, can the crystalline phases that appear in the PXRD during the thermal decomposition of compound 8 be indexed and identified?
- 6) Similarly at bottom of p8, "distinct crystalline phase", could the authors clarify if they have been able to identify or index this data?

Reviewer #3

(Remarks to the Author)

The manuscript by Knapp and co-workers describes the synthesis of Al complexes with beta-ketoiminate ligands, which are converted to hydride species and subsequently studied for their ability to deposit metallic Al at low temperature. One of the complexes shows ligand-fragmentation upon reaction with $LiAlH_4$ as a hydride source, and a Al_3 cluster is obtained with bridging imido ligands. The yield via this route is low, but the compound may also be obtained in a more straightforward manner, albeit that the yield is also moderate (34%) with this protocol.

The most significant finding is that cluster 8 is not pyrophoric, and decomposes at low temperature (major weight loss at ca. 100-150C) to form Al metal as a powder. However, this only occurs under dynamic vacuum, as otherwise some organic fragments is retained that prevents metallic Al from being formed.

- This is an interesting observation, but the manuscript and data discussed in it are preliminary in my view: especially the characterization of the product obtained under vacuum and its comparison to the material produced under N_2 atmosphere would be valuable to expand upon. How much weight loss is observed under vacuum? Is metallic Al the sole product, or is a mixture obtained? What is the organic-containing product from the synthesis under N_2 , is it still molecular in nature and which organic fragment does it contain?

- A major point is that no conductivity (resistivity) data are reported. Showing this data and how it compares to films prepared using 'conventional' pyrophoric precursors would in my view significantly strengthen the manuscript.

- While it is great that the cluster is non-pyrophoric, I think it would be good to expand the information on stability: does it decompose in air, and what products are obtained during the slow color change reported? Are these products also precursors to metallic Al? Of course eventually this will lead to Al_2O_3 , but there likely are intermediates that have unexpected stability, to explain the non-pyrophoric nature.

- The synthetic part is clearly described and compound characterization is generally fine. One note is that the elemental analysis data for the hydride species 6 and 8 is off by quite a bit: while other data seem fine and their identity is clearly established, that cannot be said for bulk purity and I think this would be good to indicate (in the SI).

- The formation of 6 is said to involve 'backbone protonation' and 'ligand reduction': I don't think this is a good way of looking at it. Rather, the product is generated from hydride transfer to the electrophilic C atom, generating a dianionic amido-enolate ligand.

- I don't think the extended structure of 6 is formed by 'aluminium atoms in one monomer binding to lithium atoms in neighboring repeating units'. It is the bridging hydrides that hold it together, and there is likely no Al-Li bond.

- 'The reaction of LiAlH₄ with protonated ligands results in markedly different product': what are these? is this based on current work, or from the literature (then please provide a reference)

Overall, this is an interesting study that is of interest to the community of inorganic / organometallic chemists and electronic materials. However, without any data on the conductivity (and comparison to traditional pyrophoric precursors, as reported by this group in the past) it is a bit too preliminary in my view.

Version 1:

Reviewer comments:

Reviewer #1

(Remarks to the Author)

Comment on manuscript entitled "The First Non-Pyrophoric Precursor for the Low Temperature Deposition of Metallic Aluminium" – Revision-1

The authors have made significant improvements to the content and descriptions of the developed chemistry. However, there are still some points that need to be addressed before final acceptance.

In my previous comments, I suggested synthesis of an analogous compound to compound 8 by directly reacting lithium aluminum hydride with various aniline derivatives. While the authors have tested the reaction with only aniline, the likelihood of obtaining the targeted molecules is relatively low using aniline. It would be beneficial for the authors to experiment with aniline derivatives, such as 2, 6-di-isopropylaniline, to improve the chances of obtaining the desired product.

Reviewer #2

(Remarks to the Author)

I can confirm that the changes made to the paper by the authors deal with all of the comments outlined in my initial review, covering the additional context to the XRD. I also think that the inclusion of the resistance data strengthens the case made for publication.

Regarding the minor issue related to the presence of a redundant. The comment "Additionally, this route to Al(0) is free of exogenous reductant, instead the 239 ligand is reduced to aid formation of metallic aluminium." Should probably read that the ligand is oxidised (as the reducing agent). The abstract still also claims that no redundant was used. If these changes are made I would be happy to recommend publication.

Reviewer #3

(Remarks to the Author)

The authors have carefully addressed all points raised by the reviewers, and the manuscript is suitable for publication.

I have two small points that I think might be useful for clarity, but if the authors disagree (or I misinterpreted their text) then that's fine.

- the added sentence "Additionally, this route to Al(0) is free of exogenous reductant, instead the ligand is reduced to aid formation of metallic aluminium" is ambiguous in my view. If Al is reduced from Al(III) in the precursor to Al(0) in the film, then something else needs to be oxidized. This is likely the organic ligand (i.e., the ligand is not reduced but oxidized in the reaction = it acts as the reducing agent)

- my comment #5 was not very clearly phrased, and although the authors have added a brief description of the likely pathway in formation of compound 6, I think that the sentence "As a result of the backbone protonation, the ligand exhibits a -2 charge" is still confusing: rather than having a proton added ('protonation'), it is the result of adding a hydride.

Version 2:

Reviewer comments:

Reviewer #1

(Remarks to the Author)

In my opinion, the authors have answered all the question raised by the reviewers by performing all the recommended reaction. So, this manuscript now be accepted for publication in nature communication.

REVIEWER COMMENTS

Reviewer #1

This manuscript has two parts. The first part of this manuscript describes the synthesis and characterization of a series of penta-coordinated Al(III) complexes coordinated to β -ketiminato (acnac) ligands with the general formula $\text{Al}(\text{R-acnac})\text{Cl}$ (1-5). All of these complexes are satisfactorily characterized. The molecular structures of these complexes are reported as expected.

In the second part, the authors reacted all these aluminium precursors with various equivalents of lithium aluminium hydride to obtain suitable aluminium hydride precursors for elemental aluminum deposition. In line with the relatively poor thermal stability of the aluminium hydride complexes, aluminium hydride complexes synthesized with relatively bulky -R [ipr and Mes] groups are isolated and characterized (6, 7, and 8). The molecular structures of complexes 6, 7, and 8 are determined using single-crystal X-ray diffraction studies. Interestingly, complex 8 is formed via the complete release of the attached β -ketiminato groups. Further, looking at the β -ketiminato ligand-free structure of complex 8, the authors developed an easier and more convenient route by a simple reaction between mesityl-aniline and lithium aluminium hydride. Finally, the authors established, or attempted to establish, that complex 8 is air-stable and can decompose below 100°C to deposit elemental aluminium. The manuscript is well written, and most of the claims are supported by experimental data.

We thank the reviewer for this kind overview of the work.

My comments:

1) The reason for the preferential use of β -ketiminato ligands is not described or justified anywhere in the manuscript. I would argue for the use of the well-known hydrazine and hydroxyl complexes of aluminium hydride for the above-stated purpose.

Thank you for the suggestion, parts of the introduction and results have now incorporated this explanation and added an additional reference (see yellow highlights). To briefly comment on why we have avoided hydrazine and hydroxyl complexes: hydrazine can be difficult to handle, which is a feature we are trying to circumvent in our search for more user-friendly precursors. For aluminium hydroxide, due to the presence of direct Al-O bonds, this would lower considerably the chances of obtaining Al metal.

2) Compound 8 is the most important molecule in this manuscript, as described by the authors, and it can be synthesized following a route (preferable) without the use of β -ketiminato ligands. Thus, the synthesis, characterization, and chemistry discussed in the major part of this manuscript have no relation to the actual goal of this work. It seems that the important molecule 8 can also be synthesized following a much longer and more tedious β -ketiminato route.

The direct synthesis approach is a powerful tool to creating compound 8 (and subsequent Al films), but lacks the versatility of the stepwise syntheses which contains

information which will prove beneficial for other chemists in the field. As such, we still believe the manuscript is stronger and more useful involving both approaches

3) I would suggest (with full respect) reporting the β -ketiminato chemistry in a different manuscript. Additionally, the authors should develop a new route for the synthesis of diverse precursors following the reaction between lithium aluminium hydride and diverse aniline/amine derivatives. Only then will the true potential of such aluminium complexes be understood in their ability to deposit metallic aluminium below 100°C. The underlying chemistry would be very important for the development of aluminium complexes for metallic aluminium deposition.

These are good points, as per our comment above, we respectfully disagree with removing the first part of the synthesis, as we believe it transparently shows a versatile synthetic route and design strategy so that others could repeat and modify. It is interesting that the reviewer has commented regarding reactions between LiAlH_4 and other aniline/amine derivatives, as we have tried these reactions without success. In light of the reviewers comments we have now incorporated a selection of these failed experiments into the SI (SI section 2.3, and reference to this in manuscript highlighted).

4) The authors have carefully used the word “non-pyrophoric” in the title of the manuscript. However, in my opinion, molecule 8 may not catch fire upon contact with air but could decompose in solution if exposed to moist air for a relatively long time. The lithium-ion coordination to the hydrogen atom associated with aluminium likely reduces the reactivity of 8 towards oxygen (I guess this may be stable under dry oxygen).

It is correct to say that compound 8 does slowly decompose if exposed to moist air, and this has been explicitly noted in the manuscript. We have additionally added an NMR study of compound 8 after exposure to air to the SI (section 6.2). In our search for more user-friendly precursors, we highly regard the fact that a compound doesn't catch fire when in contact with air, seeing that, in the case of an accident in the lab, or if we hope to scale this up to industrial settings, the risk assessment of the use of compound 8 would score much lower compared to the currently used alanes, while still delivering decompositions at low temperatures.

5) All the characterization data support the formation of aluminium metal via the decomposition of 8, but this is a one-off example. It would only be conclusive through device design and measurement of conductivity potential.

We agree with the statement, and we have now provided sheet resistivity measurements of an Al film on glass. Please refer to the corrected manuscript to see the results.

Reviewer #2

Overall, this is a succinct but detailed report on the discovery and synthesis of a novel aluminium metal precursor. It represents an excellent piece of well thought out and systematic inorganic synthesis, showcasing how an appreciation of the fundamentals of ligand design, with a careful and systematic synthesis strategy, can lead to discovery of a complex with ideal properties for a desired application. As such, I would like to strongly recommend the publication of this work in Nature Communications, (after some minor corrections).

In the introduction the authors make a clear case for the value of an air stable aluminium precursor that can be incorporated into an ink formulation for low temperature reduction to aluminium metal, and the challenge that this presents in terms of avoiding pyrophoric materials, or precursors that decompose to the more stable alumina. Although the ink formulation is the key focus for the authors, their precursors would also see potential usage in the key industrial technique of CVD.

In the results, the authors then discuss the design strategy, justifying why the system of a hydride stabilised by beta-ketoiminates is a likely candidate for the desired properties. They then report their systematic trial of different beta-ketoiminates (L) ligands, to form AlL_2Cl complexes, and then the results of the attempts to replace the chloride with a hydride by reaction with $LiAlH_4$. Comprehensive characterisation of all products by SXRD, TGA, NMR, MS and elemental analysis is provided, with a detailed discussion of the SXRD structures in the main paper, with comparison to suitable, related compounds in the literature, highlighting some unusual features of interest to main group chemists.

This careful approach to the analysis reveals the most significant finding is that 'compound 8' will decompose under vacuum to aluminium metal at temperatures from 100 to 150 C, and that this precursor is non-pyrophoric. This is a significant innovation, and the videos provided in the SI are a nice touch to highlight this difference compared to dimethyl alane.

The supplementary information provides all necessary information for the experiments to be replicated, and further and more specific analysis of each sample, including figures of the NMR IR spectra, and crystal structures confirming the details provided in the main paper.

We thank the reviewer for their kind words and recommendation.

As stated above, my recommendation is to publish, but I would like the following points and corrections to be addressed:

1) In the abstract the authors state that the aluminium complex undergoes 'reductant free' conversion. I understand that they mean there is no specific reducing agent used, but as the result is metallic aluminium, there must be a reductant to meet the rules of redox chemistry and electron bookkeeping, even if it is just the one of the ligands.

We thank the reviewer for highlighting this and have amended our statement to clarify that the ligand is reduced during formation of metallic aluminium:

"Additionally, this route to Al(0) is free of exogenous reductant, instead the ligand is reduced to aid formation of metallic aluminium."

2) There is a spelling mistake on 'ellipsoids' in figure caption 2.3) On p7 in "most likely due to the geometry constrain imposed by the presence of a fourth Al atom in the N3Al4 core" the word constrain should probably read constraint.

We have corrected this error.

4) In the conclusions should the word be losing instead of loosing?

The spelling mistakes have now been corrected, thank you for pointing this out.

5) In the supplementary information, can the crystalline phases that appear in the PXRD during the thermal decomposition of compound 8 be indexed and identified?

Please see the updated manuscript with our new results, wherein the XRD pattern of the Al on glass is entirely indexed. In reference to the original XRD in the original submission unfortunately, we have not been able to index or identify all of the phases which form during the *in situ* thermal degradation study, in part due to the high noise intrinsic to the measurement. While an interesting observation worthy of note, indicative of a complex mechanism, it does not result in metallic Al, and we do not believe it impacts the arguments of the paper.

6) Similarly at bottom of p8, "distinct crystalline phase", could the authors clarify if they have been able to identify or index this data?

Note that improved XRD data has now been incorporated, however when referring to this older data the statement has been amended to provide clarity: "multiple distinct (currently unidentified) crystalline phases between 80-100°C"

Reviewer #3

The manuscript by Knapp and co-workers describes the synthesis of Al complexes with beta-ketoiminate ligands, which are converted to hydride species and subsequently studied for their ability to deposit metallic Al at low temperature. One of the complexes shows ligand-fragmentation upon reaction with LiAlH_4 as a hydride source, and a Al_3 cluster is obtained with bridging imido ligands. The yield via this route is low, but the compound may also be obtained in a more straightforward manner, albeit that the yield is also moderate (34%) with this protocol.

The most significant finding is that cluster 8 is not pyrophoric, and decomposes at low temperature (major weight loss at ca. 100-150C) to form Al metal as a powder. However, this only occurs under dynamic vacuum, as otherwise some organic fragments is retained that prevents metallic Al from being formed.

We thank the reviewer for their comments, and can now include studies that show that dynamic vacuum is not needed if the time frame of deposition is increased (please see manuscript highlighted in yellow).

1) This is an interesting observation, but the manuscript and data discussed in it are preliminary in my view: especially the characterization of the product obtained under vacuum and its comparison to the material produced under N_2 atmosphere would be valuable to expand upon. How much weight loss is observed under vacuum? Is metallic Al the sole product, or is a mixture obtained? What is the organic-containing product from the synthesis under N_2 , is it still molecular in nature and which organic fragment does it contain?

We hope we can convince the reviewer that we have now advanced this communication from preliminary to acceptable now we have now added formation of an Al thin film on glass to the manuscript. We have now included sheet resistivity measurements in the manuscript that we believe that, coupled with PXRD is enough proof that Al metallic is formed. In response to the question about the organic product -the exact composition of the organic-containing part of the synthesis under N_2 could not be unambiguously determined as its NMR spectrum showed only a broad baseline.

2) A major point is that no conductivity (resistivity) data are reported. Showing this data and how it compares to films prepared using 'conventional' pyrophoric precursors would in my view significantly strengthen the manuscript.

We agree with the statement, and we have now added sheet resistivity measurements to the manuscript which we agree significantly strengthens the manuscript.

3) While it is great that the cluster is non-pyrophoric, I think it would be good to expand the information on stability: does it decompose in air, and what products are obtained during the slow color change reported? Are these products also precursors to metallic Al? Of course eventually this will lead to Al_2O_3 , but there likely are intermediates that have unexpected stability, to explain the non-pyrophoric nature

The cluster does slowly decompose in air and a colour change from white to ochre is observed. NMR analysis of the solid did not show any distinguishable intermediates, mostly showing a large baseline “bump”. The decomposition in oxygen has now been explicitly noted in the manuscript with characterisation of the post-decomposition product provided in the supplementary information (section 6.2).

4) The synthetic part is clearly described and compound characterization is generally fine. One note is that the elemental analysis data for the hydride species 6 and 8 is off by quite a bit: while other data seem fine and their identity is clearly established, that cannot be said for bulk purity and I think this would be good to indicate (in the SI).

This is an ongoing issue with precursors, as can be seen in the TGA the molecules are designed to breakdown under heating as low as 100°C, therefore it is not unusual for EA to be off, as there can be some loss during the combustion. We have noted this discrepancy in the methods section.

5) The formation of 6 is said to involve ‘backbone protonation’ and ‘ligand reduction’: I don’t think this is a good way of looking at it. Rather, the product is generated from hydride transfer to the electrophilic C atom, generating a dianionic amido-enolate ligand.

This is a possible reaction pathway, and we have added this as a viable route in the manuscript. We have modified this section of the manuscript to include both possible mechanisms:

“The exact formation mechanism of 6 remains unclear, although the product suggests either a step involving hydroalumination across the C–N double bond of the ligand,^{30,31} or hydride transfer to the electrophilic carbon C3 and, in turn, forming the unusual dianionic amido-enolate ligand.”

6) I don’t think the extended structure of 6 is formed by ‘aluminium atoms in one monomer binding to lithium atoms in neighboring repeating units’. It is the bridging hydrides that hold it together, and there is likely no Al-Li bond.

Our original statement was unclear, and we thank the reviewer for pointing this out. There are no Al-Li interactions in the extended structure, rather the interaction is between bridging hydrides. We have amended the statement in the manuscript to reflect this:

“The extended structure is formed through the interaction of select bridging hydride ligands with the electrophilic Al and Li sites, resulting in a zig-zag coordination polymer (Figure S2b).”

7) ‘The reaction of LiAlH₄ with protonated ligands results in markedly different product’: what are these? is this based on current work, or from the literature (then please provide a reference).

We agree this is confusing and therefore have deleted this statement.

Overall, this is an interesting study that is of interest to the community of inorganic / organometallic chemists and electronic materials. However, without any data on the conductivity (and comparison to traditional pyrophoric precursors, as reported by this group in the past) it is a bit too preliminary in my view.

We thank the reviewer for their comments, and hope that with our additional experiments revealing a highly conductive film (which has been compared to traditional pyrophoric precursors, as reported by us in the past – see yellow highlight) renders this work worthy of publication.

Response to Reviewers

REVIEWER COMMENTS

Reviewer #1

Comment on manuscript entitled “The First Non-Pyrophoric Precursor for the Low Temperature Deposition of Metallic Aluminium” – Revision-1. The authors have made significant improvements to the content and descriptions of the developed chemistry. However, there are still some points that need to be addressed before final acceptance.

In my previous comments, I suggested synthesis of an analogous compound to compound **8** by directly reacting lithium aluminium hydride with various aniline derivatives. While the authors have tested the reaction with only aniline, the likelihood of obtaining the targeted molecules is relatively low using aniline. It would be beneficial for the authors to experiment with aniline derivatives, such as 2, 6-di-isopropylaniline, to improve the chances of obtaining the desired product.

We have been investigating these reactions and have carried out the reaction with 2,6-di-isopropylaniline as the reviewer suggested. Whilst NMR does indeed suggest the reaction has worked and the product is similar to **8**, crystallisation is proving more challenging than expected, with the crystals obtained so far not being of sufficient quality to unambiguously characterise them. Please see the NMR's below, we do not agree with putting this work into the SI of the paper because it's not authenticated, a cluster structure cannot be confirmed by NMR alone.

Figure 1. Multinuclear NMR of product from 4:3 reaction of LiAlH_4 and 2,6-di-isopropylaniline.

We believe we have presented in this communication a platform reaction for the field to explore a wider array of materials. Whilst the 2,6-di-isopropylaniline analogue is expected to behave similarly to structure 8 reported here, a full series is ongoing work within the group.

Reviewer #2

I can confirm that the changes made to the paper by the authors deal with all of the comments outlined in my initial review, covering the additional context to the XRD. I also think that the inclusion of the resistance data strengthens the case made for publication.

We thank the reviewers for these comments.

Regarding the minor issue related to the presence of a redundant. The comment “Additionally, this route to Al(0) is free of exogenous reductant, instead the 239 ligand is reduced to aid formation of metallic aluminium.” Should probably read that the ligand is oxidised (as the reducing agent). The abstract still also claims that no redundant was used. If these changes are made I would be happy to recommend publication.

We thank the reviewer for spotting this, it is our mistake! We of course did mean oxidised, as it is the reducing agent, this has been fixed now (see manuscript, bottom of page 9, yellow highlight). Additionally, ‘reductant free’ has been removed from the abstract.

Reviewer #3

The authors have carefully addressed all points raised by the reviewers, and the manuscript is suitable for publication. I have two small points that I think might be useful for clarity, but if the authors disagree (or I misinterpreted their text) then that's fine.

We thank the reviewers.

- the added sentence "Additionally, this route to Al(0) is free of exogenous reductant, instead the ligand is reduced to aid formation of metallic aluminium" is ambiguous in my view. If Al is reduced from Al(III) in the precursor to Al(0) in the film, then something else needs to be oxidized. This is likely the organic ligand (i.e., the ligand is not reduced but oxidized in the reaction = it acts as the reducing agent)

See comment above, this was our mistake, we did mean oxidised, this has been fixed now (see manuscript, bottom of page 9, yellow highlight).

- my comment #5 was not very clearly phrased, and although the authors have added a brief description of the likely pathway in formation of compound 6, I think that the sentence "As a result of the backbone protonation, the ligand exhibits a -2 charge" is still confusing: rather than having a proton added ('protonation'), it is the result of adding a hydride.

We agree and have reworded to state: "the β -ketoiminate ligand has had a hydride added," (see yellow highlight on page 11).